# Interleukin-8 and Interleukin-6 Are Biomarkers of Poor Prognosis in Esophageal Squamous Cell Carcinoma

**DOI:** 10.3390/cancers15071997

**Published:** 2023-03-27

**Authors:** Paula Roberta Aguiar Pastrez, Ana Margarida Barbosa, Vânia Sammartino Mariano, Rhafaela Lima Causin, Antonio Gil Castro, Egídio Torrado, Adhemar Longatto-Filho

**Affiliations:** 1Teaching and Research Institute, and Molecular Oncology Research Center, Barretos Cancer Hospital—Pio XII Foundation, Barretos 14784-390, Brazil; 2Life and Health Sciences Research Institute (ICVS), University of Minho, 4710-057 Braga, Portugal; 3ICVS/3B’s—PT Government Associate Laboratory, 4806-909 Braga/Guimarães, Portugal; 4Anhanguera Faculty in Piracicaba, Piracicaba 13416-257, Brazil; 5Medical Laboratory of Medical Investigation (LIM) 14, Department of Pathology, Faculty of Medicine, University of São Paulo, São Carlos 13566-590, Brazil

**Keywords:** cytokines, tumor microenvironment, esophageal cancer, inflammation, Interleukin-8, Interleukin-6

## Abstract

**Simple Summary:**

Esophageal squamous cell carcinoma (SCC) is an extremely aggressive malignancy with high mortality rates. An important variable in understanding this aggressiveness seems to be in controlling the proliferation of malignant cells directly or indirectly related to systemic levels of inflammatory cytokines. Therefore, we sought to assess the usefulness of determining the most prominent cytokines in this context and their potential role as diagnostic biomarkers. For this purpose, we analyzed the levels of IL-1β, IL-6, IL-8, IL-10, TNF-α and IL-12p70 in a group of 70 patients with ESCC and 70 healthy individuals and detected increased levels of IL-1β, IL-6, IL-8 and IL-10 in patients with ESCC compared to controls. We also observed that patients with low IL-6, IL-8 had a significantly higher overall survival rate. To confirm these findings, we studied Kyse-30 and Kyse-410 cells cultured in mice and confirmed that increased growth of these cells was associated with recruitment/accumulation of intratumoral polymorphonuclear leukocytes.

**Abstract:**

Esophageal squamous cell carcinoma (ESCC) is a common type of cancer characterized by fast progression and high mortality rates, which generally implies a poor prognosis at time of diagnosis. Intricate interaction networks of cytokines produced by resident and inflammatory cells in the tumor microenvironment play crucial roles in ESCC development and metastasis, thus influencing therapy efficiency. As such, cytokines are the most prominent targets for specific therapies and prognostic parameters to predict tumor progression and aggressiveness. In this work, we examined the association between ESCC progression and the systemic levels of inflammatory cytokines to determine their usefulness as diagnostic biomarkers. We analyzed the levels of IL-1β, IL-6, IL-8, IL-10, TNF-α e IL-12p70 in a group of 70 ESCC patients and 70 healthy individuals using Cytometric Bead Array (CBA) technology. We detected increased levels of IL-1β, IL-6, IL-8, and IL-10 in ESCC patients compared to controls. However, multivariate analysis revealed that only IL8 was an independent prognostic factor for ESCC, as were the well-known risk factors: alcohol consumption, tobacco usage, and exposure to pesticides/insecticides. Importantly, patients with low IL-6, IL-8, TNM I/II, or those who underwent surgery had a significantly higher overall survival rate. We also studied cultured Kyse-30 and Kyse-410 cells in mice. We determined that the ESCC cell line Kyse-30 grew more aggressively than the Kyse-410 cell line. This enhanced growth was associated with the recruitment/accumulation of intratumoral polymorphonuclear leukocytes. In conclusion, our data suggest IL-8 as a valuable prognostic factor with potential as a biomarker for ESCC.

## 1. Introduction

Esophageal cancer (EC) is the eighth most common cancer worldwide which has the sixth worst prognosis, with a 5-year survival rate of 15–25% [1,2,3,4]. The two main histological types, esophageal squamous cell carcinoma (ESCC) and esophageal adenocarcinoma, have been spreading globally over the last three decades with some particular variations among them [5]. Indeed, while there has been a recent increase in the absolute numbers of esophageal adenocarcinomas in Western industrialized countries [6,7], the number of cases of ESCC has shown a small reduction in regions of high risk [7]. Despite this decrease, ESCC is still the most common type of EC globally, being highly prevalent in Asia while its incidence has remained relatively constant in the USA and Western Europe [7].

The development of ESCC depends on a wide variety of etiological factors that may or may not act concomitantly. Environmental factors and genetic predisposition with well-established risk factors such as alcohol intake, cigarette smoking and malnutrition play a decisive role in the progression and aggressiveness of this type of cancer [4,8]. More recently, obesity and chronic inflammation have also emerged as key factors in ESCC development and progression [9]. In this regard, several pro-inflammatory cytokines and growth factors have been identified as critical factors in ESCC progression, as their expression correlated with the clinical, pathological and survival rates of ESCC patients [9]. Therefore, while the mechanisms whereby cytokines modulate the anti-tumor immunity are not fully understood, these studies suggest that cytokines may be useful for diagnosis (detected when disease is present), prognosis (associated with disease outcome), or as predictive biomarkers (associated with drug response) of ESCC [9]. Among cytokines tested, the levels of Interleukin (IL)-1β, IL-6 and IL-8 have been associated with ESCC progression and metastasis, whereas the levels of IL-2, Interferon-gamma (IFN-γ), IL-12, and IL-18 appear to stimulate antitumor immune responses [9]. These data show that while on one hand, the cytokine network is crucial to potentiate anti-tumor immunity, on the other hand, it can also induce immune dysfunctions and contribute to tumor progression and metastasis [9,10]. Accordingly, ESCC pathogenesis has been associated with the deregulation of many cytokines and chemokines that are involved in metastasis and angiogenesis, the main cause of ESCC mortality. Indeed, cytokines such as IFN-γ, IL-27, IL-23, IL-12, and IL-2 have tumor suppressor functions, whereas pro-inflammatory cytokines, including IL-1, IL-6, and tumor necrosis factor-α, have tumor-promoting [10]. Furthermore, experimental observation demonstrated that the overexpression of IL-1RA in IL-1α expressing Kyse-410 EC cells decreased tumor cell proliferation and reduced expression of VEGF-A, a potent angiogenesis inducer. Additionally, chemokines and their receptors have been also shown to be important for EC prognosis. Indeed, CXCL12 and its receptor CXCR4 have been identified as biomarkers for ESCC and adenocarcinomas, while CXCL10, CCL4, and CCL5 have been shown to play an antitumoral role and prevent for ESCC progression [10]. Conversely, Tregs and Th17 cells recruited by tumor cell-derived chemokines CCL17 and CCL22 act by promoting EC pathogenesis [9].

The limited reliability of many traditional approaches combined with lack of specific symptoms in early stage ESCC results in most patients being diagnosed at advanced stages [4,11]. For this reason, the associations of therapies have been indicated as more beneficial than surgery alone [11]. Therefore, new biomarkers are essential for early diagnosis of ESCC. Indeed, when ESCC is diagnosed at an early stage, the 5-year overall survival rate can reach up to 80–90% [11,12,13]. Several potential biomarkers have been investigated, including p53 antibody, squamous cell cancer antigen (SCC-Ag) and carcinoembryonic antigen (CEA) [14,15]. While current evidence suggests that these and other markers may have diagnostic value, their sensitivity and specificity are not satisfactory [16]. In a recent study, the diagnostic potential of C-X-C motif ligand 8 (CXCL-8) levels in the sera of patients was compared with classical tumor markers (CEA and SCC-Ag) and the well-established marker of inflammation—C-reactive protein [14]. This study showed that the statistical parameters of CXCL-8 were higher than the classical tumor markers [14]. These data suggest that cytokines may be useful biomarkers to develop a low-cost, non-invasive and convenient method for routine diagnosis of ESCC and patient follow-up. However, given the complexity of ESCC and the pleotropic nature of cytokines, more studies are necessary to determine the potential of cytokines in the early ESCC diagnosis patient follow-up.

The aim of this work was to determine the association between ESCC progression and the systemic levels of inflammatory cytokines to define their usefulness as diagnostic biomarkers. To help in this regard, we herein determined the association between a panel of circulating cytokines and the progression of ESCC.

## 2. Materials and Methods

### 2.1. Human Samples

#### 2.1.1. Patients and Healthy Volunteers

Blood samples were collected (convenience sampling) from 70 ESCC patients referred for upper digestive endoscopy at Barretos Cancer Hospital. The control group included 70 healthy volunteers, matched for age and gender, recruited from healthy blood donors at the same hospital. Appendix A present the flowchart of inclusion of patients in the study. The complete information of both patients and healthy volunteers was previously described and published [17,18]. Diagnoses were performed by histological examination of samples obtained during endoscopy, before any treatment or primary esophageal surgery.

The ethic committees of the participating institutions approved this study. Written informed consent was obtained from all subjects.

#### 2.1.2. Blood Sample Collection and Analysis of Serum Cytokines

Peripheral venous blood (4 mL) was collected using Ethylenediamine tetraacetic acid (EDTA) tubes (BD Vacutainer, BD Biosciences, Franklin Lakes, NJ, USA) and centrifuged at 2125× *g* for 10 min at 4 °C. The supernatant (plasma) was separated, aliquoted and frozen at −80 °C until used. Blood samples from patients were collected prior to endoscopy and before treatment. Samples were identified with the hospital record (HR) number of each patient and healthy donor.

Plasma level of IL-1β, IL-6, IL-8, IL-10, tumor necrosis factor-α (TNF-α) and IL-12p70 were measured using the cytometric bead assay (BD Biosciences, San Jose, CA, USA) (#551811) following the manufacturer’s instructions, and as previously described [19,20]. Samples were acquired in a BD FACSCanto™ platform (BD Biosciences, San Jose, CA, USA) and analyzed with FACSDiva and FCAP Array™ software (BD Biosciences). The quantification of cytokines by flow cytometry was performed once and in unicate, following the manufacturer’s instructions. The samples processed as one batch.

### 2.2. Animal Model

#### 2.2.1. Cell Lines and Animals

The human esophageal squamous cancer cell lines, Kyse-30 (#94072011) and Kyse-410 (#94072023), were purchased from Sigma (St. Louis, MO, USA). Kyse-30 derived from well-differentiated invasive ESCC resected from the middle intra-thoracic esophagus of an untreated 64-year-old male. It is characterized by p53 mutation and amplification of cERB B, MYC and CYCLIN D1, and has a doubling time of 20.8 h [21]. On the other hand, Kyse-410 was established from a poorly differentiated invasive ESCC resected from the cervical esophagus of an untreated 51-year-old male. It is characterized by heparin binding growth factor (hst-1) and cyclin D overexpression, and a doubling time of approximately 45 hours [22].

Both cell lines were cultured at 37 °C in 5% CO_2_ and 95% humidity atmosphere in RPMI (#11875093) supplemented with 10% heat-inactivated fetal bovine serum (#10270-106), 1% L-glutamine (#25030-024), 1% HEPES (#15630-056), 1% sodium pyruvate (#11360-056) and 1% pennicilin/streptomycin (#15070-063) (all from Gibco (Waltham, MA, USA), Invitrogen (Waltham, MA, USA)), until reaching 80% confluence. At this stage, cells were tested and confirmed negative for mycoplasma, and frozen in 1 ml aliquots in liquid nitrogen. Seven days before an animal experiment, one aliquot of each cell line was thawed and grown for 1 day in a T25 flask. Cells were then cultured in T75 flasks and passed every 48 h until injected in mice.

NOD.Cg-Prkdc^scid^ Il2rg^tm1Wjl^/SzJ (NSG) mice were originally obtained from the Jackson Laboratory (stock number: 005557, https://www.jax.org/strain/005557 (accessed on 9 October 2020)) (Bar Harbor, ME, USA) and maintained in our animal facility. All mice used in this study were age- and sex-matched and between the ages of 8 and 12 weeks. Male NSG mice were injected subcutaneously with 1 × 10^6^ Kyse-30 or Kyse-410 cells in the left hind flank. Tumor growth was measured using a caliper and the volume was calculated using the following formula: (πxd^2^ × D)/6 × 1000, d = small diameter and D = large diameter. Mice were sacrificed by CO_2_ inhalation and the tumors dissected and processed for RNA extraction, flow cytometry and histological analysis. 

All animal experiments were performed according to recommendations of the European Union Directive 2010/63/EU and were previously approved by the Subcomissão de Ética para as Ciências da Vida e da Saúde (SECVS 074/2016) and the Portuguese National Authority Direcção Geral de Alimentação e Veterinária (014072).

#### 2.2.2. Flow Cytometry Analysis

Aseptically excised tumors were sectioned and incubated at 37 °C for 30 min with collagenase D (#11088866001, 0.7 mg/mL, Sigma) and disrupted into single-cell suspensions by passage through a 70-μm-nylon cell strainer (#352350, BD Biosciences). Tumor single-cell suspension was then treated with erythrocyte lysis buffer (0.87% of NH_4_Cl). In order to remove cell debris, tumor single-cell suspension was further subjected to a 40:80% Percoll gradient (#17089101, GE Healthcare, Chicago, IL, USA). The resulting cell suspension was washed twice and counted. For flow cytometry analysis, single-cell suspension was stained with fluorochrome-conjugated antibodies for 30 min on ice. Antibodies specific for CD11b (#101216, clone M1/70; dilution 1:100) and Ly6G (#127645, clone 1A8; dilution 1:100) were obtained from BioLegend (San Diego, CA, USA). Data were acquired on a LSRII flow cytometer (BD Biosciences) with Diva Software and analyzed using FlowJo software (BD Biosciences). The total number of cells was determined based on the percentage of cells determined by flow cytometry and the total number of cells counted.

#### 2.2.3. Real-Time RT-PCR

Total RNA was extracted using Triple XTractor (GB023.0100, Grisp, Porto, Portugal) according to the manufacturer’s instructions. cDNA was generated from 1 μg of RNA using the GRS cDNA Synthesis Master Mix (GK81.0100, Grisp) following the manufacturer’s instructions. The resultant cDNA template was used to quantify the expression of target genes by real-time PCR (Bio-Rad CFX96 Real-Time System with C1000 Thermal Cycler), and normalized to Ubiquitin mRNA levels using the ΔCt method (1.8^(Housekeeping gene mRNA expression—Target gene mRNA expression) × 100,000). Target gene mRNA expression was quantified using SYBR Green (Thermo Fisher Scientific, Waltham, MA, USA) and specific oligonucleotides (Invitrogen).

## 3. Statistical Analyses

The association of sociodemographic and habits between groups was performed by Chi-square test. The comparison of cytokine levels between the groups was verified by Mann–Whitney test, and the normality of data was analyzed by Kolmogorov–Smirnov test. The Odds Ratio (OR) was estimated by logistic regression in the context of multivariate analysis.

Cytokine dosages were initially treated as quantitative variables to compare the case and control groups. The ROC curve was used to determine the cut off for cytokine values in order to discriminate patients who died among the case group. In survival analyzes, cytokines were treated as qualitative variables. Kaplan–Meier and the comparison between the curves by log-rank test estimated the survival curve. To select the characteristics for multivariate analysis, we adopted a significance level of 20%. To adjust the model, only individuals who had information on all variables (*n* = 56) were considered, so no specific treatment for missing data was performed. For the selection of variables, we applied the Backward Stepwise method, considering a significance level of 5% to adjust the final model. Despite the sample size of 56 participants, the model presented a power greater than 0.90 when we considered IL12, IL8 or TNF as variables of interest (Appendix A).

To verify the assumption of proportional hazards in the Cox regression model, we used a descriptive method that consists of estimating ${\hat{\Lambda }}_{0}\left(t\right)$ according to the expression below:$${\hat{\Lambda }}_{0}\left(t\right)=\underset{j:tj<t}{\sum }\frac{dj}{\sum_{ɩ\in R_{j}}^{}\;\exp\{X_{l}^{\prime }\hat{\beta }\}}$$
where $d_{j}$ is the number of events in each of the strata in $t_{j}$. If the assumption of proportionality is valid, the log ((${\hat{\Lambda }}_{0}\left(t\right))$ versus t graph is expected to show approximately constant differences over time. In addition, we calculated Pearson’s correlation coefficient between times and standardized Schoenfeld residuals for each of the covariates. Coefficients close to zero are expected to show no evidence to reject the assumption of proportional risks. In this way, we believe that the assumption of proportional risks is valid since the log ${\hat{\Lambda }}_{0}\left(t\right))\;$versus t graphs, for each covariate, present constant distance over time and Appendix A shows that Pearson’s correlation coefficients do not display a significant difference (*p* > 0.05). These analyzes carried out with samples from humans were performed with IBM SPSS Statistic software 21.0 (SPSS, Chicago, IL, USA). *p* values ≤0.05 were considered statistically significant.

For the analysis of samples from the animal model, the data were represented as mean ± standard error of the mean (SEM). Two modes of ANOVA with multiple post-test Bonferroni comparisons and Student’s *t*-test were used for statistical comparisons. *p* values ≤ 0.05 were considered statistically significant.

## 4. Results

### 4.1. Characterization of the Study Population and Risk Factors to Develop Disease

When compared with the control group, the case group had a higher frequency of individuals living in rural areas (*p* = 0.007) that consume more alcohol in the past (*p* < 0.001) and tobacco nowadays (*p* < 0.001). Multivariate analysis of the significant variables was performed, and a significance level of α = 0.20 was adopted. All the statically significant variables obtained in the univariate analysis, with a *p* < 0.20, were included. The multivariate analysis performed included degree of education, place of residence, alcohol consumption, smoking, exposure to pesticide or insecticide, physical activity (Appendix A). The majority of the case group was also more exposed to pesticides and insecticides (*p* < 0.001), and did not perform physical activity (*p* = 0.024) (Appendix A).

This analysis also estimated the OR of those with a risk factor to develop the disease and the odds of those without a risk factor to develop the disease. We determined that only alcohol consumption, tobacco, exposure to pesticides or insecticides remained as statistically significant variables (*p* < 0.05, Table 1. Specific to alcohol consumption, individuals who drank in the past (and stopped more than 12 months ago) have a 14.5-fold increased chance of developing the disease in relation to those who never drank (95% Confidence interval (CI): 1.3–163.6, *p* = 0.030). Regarding the use of tobacco, individuals who currently smoke have a 11.3-fold increased chance of developing esophageal cancer compared to those who have never smoked (95% CI: 2.9–44.3, *p* < 0.001). Individuals exposed to pesticides and insecticides displayed results of 3.2-fold increased chance of developing the disease compared to those who were not exposed (95% CI: 1.1–9.4, *p* = 0.033). Taken together, these data corroborate alcohol consumption and tobacco usage as risk factors for ESCC, according to International Agency for Research on Cancer [23].

Regarding the clinicopathological features, 57.4% of the tumors were localized in the middle third of the esophagus, 53.8% were moderately differentiated, and 71.2% were classified with a TNM staging of III and IV [24] (Appendix A).

### 4.2. Serum Cytokine Analysis Reveals a Subset of Cytokines Differentially Produced by ESCC Patients

From the initial set of 6 cytokines, we determined higher levels of IL-1β (*p* = 0.032), IL-6 (*p* < 0.001), IL-8 (*p* < 0.001), and IL-10 (*p* = 0.027) in ECSS patients than in controls (Table 2). Multivariate analysis, including the cytokines IL-1β, IL-6, IL-8, and IL-10 were significantly altered between cases and controls (Table 2), revealed that only IL-8 remained statistically significant (OR: 1.6–95% CI: 1.3–2.0, *p* < 0.001) (Table 1). The data that emerged from multivariate analysis suggest that high systemic level of IL-8, but not IL-6, is associated with ESCC progression, despite the fact that IL-6 was highly significant for poor overall survival estimate by Kaplan–Meier analysis (Table 3).

The relationship between the levels of the different cytokines and survival was established using the cutoff point obtained with the ROC curve (Appendix A); based on these data, individuals were categorized into two distinct groups: low and high cytokine producers (Appendix A). In addition to cytokines, in this analysis, we also included the socio-demographic characteristics, lifestyle and clinical-pathological data of our cohort. The time of overall survival was calculated as the interval between the date of diagnosis and the date of death from cancer or the date of the last information. Kaplan–Meier curves are represented in Appendix A. Using the Kaplan–Meier analyses and log-rank test, we observed that only IL-6, IL-8, surgery and TNM [24] staging had significant influence on the overall survival rate at 12, 36 and 60 months (Table 3). Individuals who underwent surgery had significantly better overall survival rate (90.0% at 12 months, 80.0% at 36 months and 51.3% at 60 months) when compared to individuals who did not undergo surgery (47.1% at 12 months, 16.5% at 36 months and 8.2% at 60 months). Additionally, individuals with TNM [24] I/II staging at the time of diagnosis displayed significantly better overall survival rates (87.8% at 12 months, 81.6% at 36 months and 49% at 60 months) than individuals with advanced TNM [24] staging of III/IV (52.4% at 12 months, 16.7% at 6 months and 9.5% at 60 months).

Regarding the systemic level cytokines, we determined that individuals with low levels of IL-6 had significantly higher overall survival rate (80.4% at 12 months, 64.3% at 36 months and 45.1% at 60 months) when compared to individuals with high levels of IL-6 (48.8% at 12 months, 18.2% at 36 months and 6.9% at 60 months). Similar data were obtained for IL-8, wherein patients with low levels of this cytokines had notably higher overall survival rates (89.1% in 12 months, 50.4% in 36 months and 41.2% in 60 months) than patients with high levels of IL-8 (41.5% at 12 months, 25.9% at 36 months and 9.3% at 60 months). Overall survival rates correlating IL-6, IL-8, TNM and Surgery are shown in Figure 1. These data show that the pro-inflammatory cytokines IL-6 and IL-8 are closely related to tumor growth and progression. In this regard, these cytokines have been shown to promote immune escape, epithelial–mesenchymal transition, and recruitment of myeloid-derived suppressor cells thus being associated with poor prognosis in many malignant tumors [25,26]. To identify the clinical value of the tested cytokines in cancer diagnosis, the area under the curve (AUC) was calculated (Appendix A). For each cytokine, we established an AUC > 0.70 (or 70%). AUC of IL-8 (76%) was the highest of all the cytokines together with IL-6. The specificity was also the highest of all cytokines (71%), whereas the sensitivity was the lowest (73.3%). IL-8 showed high sensitivity correctly discriminating individuals with cancer.

Multivariate analysis using COX regression was performed to calculate the Hazard Ratio estimates (HR) (95% confidence intervals (CI)) for the significant variables (*p* < 0.005) (Table 4). Individuals who have high levels of IL-12p70 have a lower risk of death when compared to those with lower levels of this cytokine (HR: 0.34–95% CI: 0.14–0.79). Similarly, individuals with high level of TNF-α have a lower risk of death when compared to those with low TNF-α levels (HR: 0.23–95% CI: 0.09–0.59). In contrast to IL-12p70 and TNF-α, IL-8 appears to be a risk factor for ESCC, since individuals with higher levels of IL-8 had increased risk of dying than individuals with low levels of IL-8 (HR: 4.56–95% CI: 2.21–9.41). As expected, patients who underwent surgery and radiotherapy had lower risk of dying when compared to those who did not undergo surgery or did not undergo radiotherapy, (HR: 0.21–95% CI: 0.08–0.55) and (HR: 0.27–95% CI: 0.14–0.54), respectively. Patients with early-stage TNM [24] (I/II) also had a lower risk of dying when compared to those with advanced TNM [24] (III/IV) (HR: 3.68–95% CI: 1.45–9.37) (Table 4).

### 4.3. The Enhanced Growth of ESCC Cell Lines in Mice Associates with Increased Tumor Neutrophil Infiltration

Representative haematoxylin and eosin of histological section tumors induced by Kyse-30 and Kyse-410 are depicted in Figure 2A–D. 

IL-8 (CXCL-8) was originally described as a chemoattractant of polymorphonuclear leukocytes by acting on CXCR1/2 [27]. Recent data suggest that this chemokine can exert pro-tumoral functions, by stimulating angiogenesis, promoting survival of cancer stem cells and recruiting immunosuppressive myeloid cells [27]. To further address the role of IL-8 in ESCC, we developed a mouse model using esophageal cancer lines. As there is no homologue IL-8 in mice, CXCR2 mediates neutrophil chemotaxis in response to human IL-8 and to the murine CXCL1, CXCL2, and GCP-2/liposaccharide-induced CXC chemokine (LIX) [28,29]. We measured the expression of CXCL-1 and CXCL-2 in the tumor tissue and detected higher expression of CXCL-1 in Kyse-410 than Kyse-30 tumors while the expression of CXCL-2 was below the detection limit for both tumors (Figure 1 and Appendix A). As CXCL-1 plays an important role in the recruitment of polymorphonuclear leukocytes to inflammatory sites, we next determined the accumulation of these cells (CD11b+ Ly6G+) (Figure 2E). As the murine orthologs of human GRO-α (CXCL1) and GRO-β (CXCL2) are often studied instead of IL-8, we determined the expression of these chemokines in ESCC tumor bearing mice. Esophageal lines Kyse-30 or Kyse-410 were subcutaneously injected in mice and follow tumor growth over a period of 20 days. These two ESCC cell lines were selected because they display different aggressiveness in mice. The Kyse-410 cell line is considered to be of lower aggressiveness than Kyse-30 cell line because it has higher expression of miR-644a previously demonstrated to inhibit ESCC cell growth, migration, and invasion in vitro and suppressed tumor growth and metastasis in vivo [30]. We determined that animals injected with Kyse-30 displayed larger tumors when compared to animals injected with the Kyse-410 (Figure 2E). Flow cytometry analysis revealed a significant increased intratumor accumulation of polymorphonuclear leukocytes in Kyse-30 than in Kyse-410 tumors (*p* = 0.0367) (Figure 2F). A representative scheme depicting the gating strategy used for the analysis of intratumoral polymorphonuclear leukocytes in Kyse-30 or Kyse-410 is provided in Appendix A. These data suggest that the enhanced growth of ESCC cell lines in mice model is associated with the recruitment/accumulation of intratumoral polymorphonuclear leukocytes. Evaluation of the intra-tumor immune cell infiltrate in the tumors induced by the Kyse-30 and Kyse-410 cell lines by flow cytometry are represented in Figure 2G.

## 5. Discussion

Over the recent years, several proteins including cytokines and chemokines have been proposed as biomarkers for esophageal cancer diagnosis. However, the role of the cytokine/chemokine network in the context of cancer is complex. While these small molecules play crucial roles in orchestrating the anti-tumor immune response, emerging evidence suggests that the cytokine and chemokine network can be co-opted to promote tumor growth and aggressiveness.

The elevated mortality rates caused by esophageal cancer are associated with its rapid progression and late-stage diagnosis [31]. More than half of the esophageal cancer cases are diagnosed in advanced stages; indeed, the absence of specific symptoms delays diagnosis through endoscopic ultrasonography or computed tomography [32]. As such, this delay results in the presence of metastases and consequently poor prognosis. Therefore, the identification of novel biomarkers that can help in the diagnostic of early stages of the disease is crucial to curb the mortality of this cancer worldwide [33]. We evaluated a panel of cytokines and chemokines in ESCC patients to determine their potential as biomarkers for the diagnosis and progression of ESCC. We showed that IL-1β, IL-6, IL-8 and IL-10 were significantly increased in the plasma of ESCC patients when compared to controls. However, after multivariate analysis, only IL-8 was an independent predictive factor of ESCC prognosis, together with other well-known risk factors such as alcohol, tobacco and pesticide/insecticide exposure [34]. The association of IL-8 with development/progression of ESCC has been previously suggested [35,36,37,38,39,40]. Despite this evidence, the mechanisms whereby IL-8 is involved in ESCC progression are not completely known [37,38,39,40]. In this regard, IL-8 has been shown to act as neutrophil chemoattractant [41,42]. Interestingly, neutrophils can adopt an immunosuppressive phenotype in the context of tumors by producing chemokines that promote the recruitment of regulatory T cells (Tregs) [43], and by synthetizing molecules involved in invasion and metastasis such as matrix metalloproteinases (MMPs) and vascular endothelial growth factor (VEGF) [44,45]. Our mouse data is in concordance with this pro-tumor role of neutrophils, as the more aggressive growth of ESCC cells was associated with enhanced neutrophil infiltration. However, while it is tempting to speculate that the increased infiltration of neutrophils is the cause for enhanced tumor growth, more data are required to clarify the role of neutrophils in ESCC progression.

We also unveiled an association between the levels of IL-6 and IL-8 with the overall survival rate of patients with ESCC. Patients with ESCC that had high plasma levels of IL-6 or IL-8 also displayed reduced survival rste when compared to patients with low levels of these cytokines. This reduced survival rate was also seen for patients diagnosed with TNM III and VI and patients that did not undergo surgery. These results can be explained by the pro-inflammatory functions of these cytokines in addition to stimulating angiogenesis, invasion, and metastasis [46,47,48]. Specifically, IL-6 not only plays a key role in inducing acute phase inflammation and initiating the innate immune response [49], but also stimulates malignant transformation, tumor progression and cachexia associated with several tumor types [50]. In addition, IL-6 can directly stimulate the expression of VEGF and promote angiogenesis and MMPs, which play a fundamental role in tumor invasion and extracellular matrix differentiation [48]. In line with these observations, it has been shown that patients with elevated IL-6 levels display enhanced tumor progression, poor prognosis and poorer overall survival rates [48,51]. IL-8 coordinates multiple functions, from recruiting neutrophils and myeloid-derived suppressor cells to inhibiting apoptosis and enhancing angiogenesis, epithelial–mesenchymal transition and tumor cell growth [48,52]. Circulating IL-8 was elevated in patients with ESCC and positively correlated with the presence of lymph node and distant metastases as well as with inflammatory status of cancer patients [36], suggesting IL-8 as a biomarker for ESCC [14,35]. However, despite the significant difference in the levels of several cytokines identified in patients with ESCC, our study shows that one key limitation of cytokines for diagnosis/prognosis is their relatively low sensitivity and specificity (including IL-8) [10].

## 6. Conclusions

Both interleukins, IL-8 and IL-6, showed promising results to identify parameters of greater aggressiveness of the biological behavior of ESCC. IL-8 in particular is a promising valuable prognostic factor to be used as biomarker for ESCC because of its significant relationship with metastases. IL8 was also significant by multivariate analysis (where only variables with *p* < 0.2 were included). However, despite its exploratory and unbiased nature, our study presents limitations. Specifically, while our data further cement the involvement of IL-8 in ESCC development/progression, we are still unable to determine the role of cytokine levels in tumor progression. Moreover, our approach did not unravel a temporal role of IL-8 in the development/progression of ESCC. Another limitation is the small number of samples. Further studies with a larger sample size may help to clarify the real role of cytokines, especially IL-8, in the development of esophageal cancer.

## Figures and Tables

**Figure 1 cancers-15-01997-f001:**
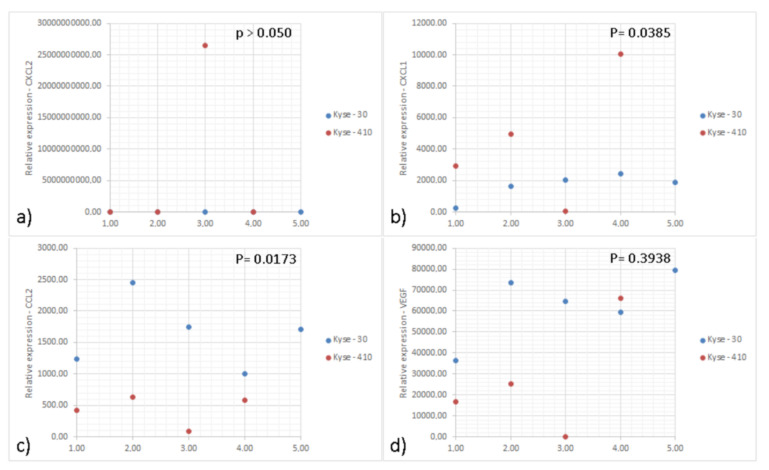
Graphs of relative expression of genes of interest performed by real-time PCR. Graphs representing the relative expression of the genes (**a**) CXCL-2; (**b**) CXCL-1; (**c**) CCL2; and (**d**) VEGF in the tumors induced by the Kyse-30 and Kyse-410 cell lines. “BDL: Below Detection Levels”. Data represent 10 animals (5 per group) from the same experiment. Each dot represents one animal; the blue dots indicate the relative expression of the different genes of each animal inoculated with Kyse-30 and the red dots of each animal inoculated with Kyse-410. All experiments were performed only once. Statistical significance was calculated by using unpaired *t*-test and *p* values are shown in the figure.

**Figure 2 cancers-15-01997-f002:**
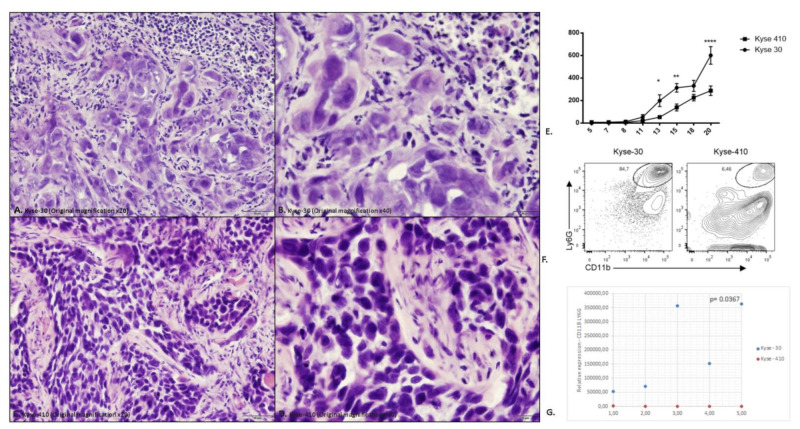
Evaluation of the intra-tumor immune cell infiltrate in the tumors induced by the Kyse-30 and Kyse-410 cell lines. (**A**) Representative haematoxylin and eosin (H&E) of histological section tumors induced by Kyse-30 (Original magnification ×20); (**B**) Representative H&E of histological sections tumors induced by Kyse-30 (Original magnification ×40); (**C**) Representative H&E of histological sections tumors induced by Kyse-410 (Original magnification ×20); (**D**) Representative H&E of histological sections tumors induced by Kyse-410 (Original magnification ×40). (**E**) Tumor growth curve in NSG mice inoculated with the Kyse-30 and Kyse-410 cell lines. Data represent 10 animals (5 per group) from the same experiment. Each dot represents one animal. The square dots represent each animal inoculated with Kyse-30 cell line and the spherical dots represent each animal inoculated with Kyse-410 cell line. The y-axis represents the tumor volume and the x-axis the days after inoculation of the cells. This experiment was performed only once. Statistical significance was calculated by using Anova test (* *p* = 0.024; ** *p* = 0.007; **** *p* < 0.0001). (**F**) Representative flow cytometry data of intratumoral polymorphonuclear leukocytes (CD11b+ Ly6G+) in Kyse-30 or Kyse-410 tumor-bearing mice. (**G**) Evaluation of the intra-tumor immune cell infiltrate in the tumors induced by the Kyse-30 and Kyse-410 cell lines by flow cytometry. Data represent 10 animals (5 per group) from the same experiment. Each dot represents one animal; the blue dots indicate CD11b+Ly6G+ expression of each animal inoculated with Kyse-30 and the red dots indicate CD11b+Ly6G+ expression of each animal inoculated with Kyse-410. All experiments were performed only once. Statistical significance was calculated by using unpaired *t*-test (*p* = 0.0367).

**Table 1 cancers-15-01997-t001:** Multiple logistic regression model of the statistically significant variables between the groups (case/control).

Variable/Category	N Cases (N Events)	OR (CI95%)	*p*
**Alcohol**			
Never	15 (3)	1	-
Yes, currently (if it was stopped in the last 12 months)	85 (36)	3.9 (0.4–35.9)	0.227
Yes, in the past	40 (31)	14.5 (1.3–163.6)	0.030
**Tobacco**			
Never	47 (10)	1	-
Yes, currently (if it was stopped in the last 12 months)	58 (47)	11.3 (2.9–44.3)	<0.001
Yes, in the past	35 (13)	1.5 (0.3–6.6)	0.581
**Exposure to pesticide or insecticide**			
No	78 (28)	1	-
Yes	60 (42)	3.2 (1.1–9.4)	0.033
**IL-8**	140 (70)	1.6 (1.3–2.0)	<0.001

N: number; OR: Odds Ratio; CI 95%: Confidence interval at the level of 95%; IL: Interleukin. Utilized test: Wald test. Statistically significant if *p* < 0.05.

**Table 2 cancers-15-01997-t002:** Comparison of the plasma levels of cytokines between study groups (case/control).

Cytokines	Groups				
	Case		Control		*p*
	Mean (SD) pg/mL	Median (Min–Max) pg/mL	Mean (SD) pg/mL	Median (Min–Max) pg/mL	
IL-12p70	0.78 (0.99)	0.39 (0.00–4.83)	0.93 (1.61)	0.28 (0.00–11.51)	0.991
TNF-α	0.95 (2.24)	0.05 (0.00–16.37)	0.86 (2.14)	0.05 (0.00–14.27)	0.772
IL-10	12.98 (97.53)	0.68 (0.00–816.59)	0.55 (0.78)	0.42 (0.00–5.73)	0.027
IL-6	1157.20 (9614.02)	6.72 (0.21–80446.68)	2.07 (2.49)	1.62 (0.00–18.18)	<0.001
IL-1β	1.21 (3.07)	0.49 (0.00–24.60)	0.66 (1.87)	0.00 (0.00–13.76)	0.032
IL-8	196.31 (1529.42)	7.51 (2.40–12806.73)	3.52 (3.05)	3.16 (0.64–25.59)	<0.001
Total	70		70		

SD: standard deviation; pg/mL: picograms per milliliter; IL: Interleukin; TNF-α: Tumor necrosis factor α. Utilized test: Mann–Whitney test. Statistically significant if *p* <0.05.

**Table 3 cancers-15-01997-t003:** Overall survival estimated by Kaplan–Meier method considering socio-demographic, lifestyle, clinicopathological data and plasma levels of cytokine variables.

Variable/Category	n Cases (n Events)	% Probability of Survival			*p*
		12 Months	36 Months	60 Months	
**Overall survival**	70 (52)	60.8	36.0	21.5	-
**Sex**					
Female	11 (7)	60.0	50.0	18.8	0.403
Male	59 (45)	60.7	33.2	21.1	
**Race**					
White	50 (38)	64.0	35.1	23.6	0.208
Non-white	18 (14)	48.1	34.4	0.00	
**Exposure to pesticide or insecticide**					
No	28 (21)	64.3	46.4	26.7	0.360
Yes	42 (31)	58.6	28.0	17.4	
**Place residence**					
Urban area only	12 (8)	66.7	41.7	41.7	0.553
Rural area only	8 (7)	62.5	25.0	12.5	
Both areas	50 (37)	59.1	36.5	17.7	
**Alcohol**					
Never/In the past	34 (26)	58.0	36.7	17.8	0.839
Yes, currently (if it was stopped in the last 12 months)	36 (26)	63.5	35.1	24.6	
**Tobacco**					
Never	10 (8)	30.0	20.0	20.0	0.315
Yes, currently (if it was stopped in the last 12 months)	47 (35)	60.8	39.1	20.3	
Yes, in the past	13 (9)	84.6	38.5	28.8	
**Degree of differentiation**					
Well-differentiated	9 (7)	55.6	22.2	22.2	0.807
Moderately differentiated	35 (26)	56.4	37.6	23.9	
Little-differentiated	21 (16)	66.0	30.5	16.9	
**TNF-α**					
<0.825	48 (39)	58.3	25.9	18.9	0.100
≥0.825	22 (13)	66.7	61.2	28.5	
**IL-10**					
<1.34	50 (38)	59.1	29.6	22.0	0.470
≥1.34	20 (14)	65.0	53.2	19.7	
**IL-6**					
<4.7	27 (14)	80.4	64.3	45.1	**<0.001**
≥4.7	43 (38)	48.8	18.2	6.9	
**IL-1β**					
<1.35	52 (40)	62.6	29.2	19.6	0.364
≥1.35	18 (12)	55.6	55.6	28.8	
**IL-8**					
<6.75	29 (16)	89.1	50.4	41.2	**0.001**
≥6.75	41 (36)	41.5	25.9	9.3	
**Surgery**					
No	44 (40)	47.1	16.5	8.2	**<0.001**
Yes	20 (9)	90.0	80.0	51.3	
**Radiotherapy**					
No	25 (22)	56.0	28.0	7.0	0.117
Yes	43 (29)	61.8	41.2	29.7	
**Chemotherapy**					
No	24 (20)	41.7	33.3	15.6	0.186
Yes	45 (32)	70.3	36.3	24.6	
**TNM**					
I/II	17 (8)	87.8	81.6	49.8	**<0.001**
III/IV	42 (38)	52.4	16.7	9.5	
**Topography of the tumor**					
Upper/Middle third	33 (27)	69.7	42.4	20.5	0.447
SOE	19 (14)	51.3	22.8	22.8	

n: Number; IL: Interleukin; TNF-α: Tumor necrosis factor α; TNM staging: System based on the size and/or extent of the primary tumor (T), amount of compromised lymph nodes (N) and presence of metastases (M). Utilized test: Log-rank. Statistically significant if *p* < 0.05.

**Table 4 cancers-15-01997-t004:** Hazard Ratio estimation by the multiple Cox model for the overall survival time in esophageal cancer individuals.

Variable/Category	n Cases (n Events)	HR (IC95%)	*p*
**Race**			
White	43 (34)	-	-
Non-white	13 (11)	5.99 (2.28–15.72)	<0.001
**IL-12p70**			
<1.23	40 (33)	-	-
≥1.23	16 (12)	0.33 (0.15–0.73)	0.007
**TNF-α**			
<0.825	41 (36)	-	-
≥0.825	15 (9)	0.23 (0.09–0.59)	0.003
**IL-8**			
<6.75	22 (14)	-	-
≥6.75	34 (31)	4.56 (2.21–9.41)	0.002
**Surgery**			
No	40 (37)	-	-
Yes	16 (8)	0.21 (0.08–0.55)	0.002
**Radiotherapy**			
No	25 (22)	-	
Yes	31 (23)	0.27 (0.14–0.54)	<0.001
**TNM**			
I/II	16 (8)	-	
III/IV	40 (37)	3.68 (1.45–9.37)	0.006

n: Number; HR: Adjusted Hazard Ratio for smoking; IC 95 Confidence interval at the level of 95%; IL: Interleukin. Utilized test: Cox regression. Statistically significant if *p* < 0.05.

## Data Availability

The data that support the findings of this study are available from the corresponding author upon reasonable request. Written informed consent was obtained from all subjects.

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
