# Peer review of "Interleukin-8 and Interleukin-6 Are Biomarkers of Poor Prognosis in Esophageal Squamous Cell Carcinoma"

_cancers, 2023, doi:10.3390/cancers15071997_

Round 1

Reviewer 1 Report

1. After multifactor analysis, different treatment schemes (surgery, radiotherapy and chemotherapy, etc.) among the enrolled patients are the factors that affect the OS of patients. How can this explain that IL-8 is an independent influencing factor?

2. In the plasma test, both IL-8 and IL-6 are abnormal and related to the patient's OS, but why only select IL-8 for the test (mentioned slightly in this discussion, but I don't think it is clear)?

3. It is recommended to analyze the effects of IL-8, IL-6 and other factors in the surgical and non-surgical subgroups

Author Response

Response to Reviewers

Reviewer 1

  1. After multifactor analysis, different treatment schemes (surgery, radiotherapy and chemotherapy, etc.) among the enrolled patients are the factors that affect the OS of patients. How can this explain that IL-8 is an independent influencing factor?

Thank you for questioning this claim regarding IL-8. As circulating IL-8 was elevated in patients with ESCC and positively correlated with the presence of lymph node and distant metastases, we classified this IL as more important to define worse behaviour (including metasis).  We discussed our results and modified the text, as well as the Title of the work, to avoid inappropriate conclusions and we also highlight the role of IL-6 as a promising biomarker for worse prognosis of oesophageal carcinoma. New Title: Interleukin-8 and Interleukin-6 are biomarkers of poor prognosis in esophageal squamous cell carcinoma.

  1. In the plasma test, both IL-8 and IL-6 are abnormal and related to the patient's OS, but why only select IL-8 for the test (mentioned slightly in this discussion, but I don't think it is clear)?

When analyzing the expression of cytokines in relation to survival, we found that individuals with high levels of cytokines IL-6 and IL-8 had worse survival at 12, 36 and 60 months, when compared to individuals with plasma levels of these cytokines. However, subsequently, the variables that defined a value of p <0.2 were selected to compose the multivariate analysis using COX Regression (Table 4). Thus, only the IL-8 cytokine was statistically significantly, appearing to be a risk factor for the disease, since individuals who had increased levels of the cytokine had a greater risk of dying when compared to individuals with low levels of IL-8. 8 (HRaj: 4.56 - 95% CI: 2.21 - 9.41).

  1. It is recommended to analyse the effects of IL-8, IL-6 and other factors in the surgical and non-surgical subgroups. We improved Discussion with focus in both ILs, as described below:

“Both interleukins, IL-8 and IL-6, showed promising results to identify parameters of greater aggressiveness of the biological behaviour of ESCC. IL-8, particularly, is a promising valuable prognostic factor to be used as biomarker for ESCC because of its significant relationship with metastases. IL8 was also significant by multivariate analysis (where only variables with p<0.2 were included).”

Reviewer 2 Report

Review of Pastrez et al

            The authors have evaluated the levels of various cytokines in the plasma as well as various socioeconomic and behavioral factors, stage of cancer and cancer and life outcome in a group of  70 patients with esophageal squamous cell carcinoma and compared these to 70 normal volunteers. They also extend their studies of ESCC to a mouse model using two ESCC lines with different degrees of tumor proliferation. Overall, the study is well designed and the investigations carried out well. The data collection is complete and well edscribed. However, there are several substantive issues that must be rectified.

1.      The biggest issue is conclusion by the authors that IL-8 levels, and IL-8 levels alone, are “independent” markers of disease progression and outcome. Their data show that IL-6 is also strong marker of disease progression and outcome. Their entire claim about IL-8 rests on statistical jugglery in Table 2 – something they call “multivariate analysis.” I require the numerical data for IL-6 to be included in Table 2. Also I require the authors to note that their own data show that IL-6 is singularly important (see Table 3). Also see the non-existent Kaplan Meir plot that should be Fig. 1. (its missing). Also please show IL-6 in Table 4. What emerges is that ignoring the statistical jugglery – both IL-6 and IL-8 are informative. Thus the authors must revise the MS to highlight IL-6 (in title, abstract and keywords).

2.      The second major problem is that various data items are missing. There is no Kaplan-Meir Figure 1. The various tables (especially in Supplement) are not in serial order as in the text. Fig. 2E in text is not in serial order with respect to Fig. 1. This makes it difficult for a reader to follow what the authors are saying.

3.       There are many language issues: these are enumerated below.

25  delete “causisstry” and rephrase with “group”

27  delete “than” “compared to”

36  in keywords add IL-6

45 delete “slightly” What does this mean? Rephrase

45  “decrease” What decrease? Previous sentence said increase and also slightly decrease. Rephrase.

48. Rephrase “The etiology of ESCC is complex” wherein …

52 delete “a”

52  its “Bhat”

53 delete “report”

54 change are to were – past tense

55 correklated past tense

59  change “in” to “on the”

60 change “in” to “on”

70 observations  plural

72 inducer

74 delete the

75 shown to be biomarker

84 change have to has

89 sduggests singular

109  what does “stipulated” mean. In Fig. S1 delete “included for convenience” Ha!!

145 thawed past tense

163 suspensions plural

226 out of serial order Table S3

229  Table S4 not in serial order

241 change what to to

273 In Table 2 include IL-6 values

297-299 there is no Fig. 1 Kaplan Meir plot

Please show IL-6 in Table 4

355 and 358 Fig 2E and Fig 1 are not in serial order

Ref. 21 is missing information

Author Response

Response to Reviewers

Reviewer 2

Review of Pastrez et al

            The authors have evaluated the levels of various cytokines in the plasma as well as various socioeconomic and behavioral factors, stage of cancer and cancer and life outcome in a group of  70 patients with esophageal squamous cell carcinoma and compared these to 70 normal volunteers. They also extend their studies of ESCC to a mouse model using two ESCC lines with different degrees of tumor proliferation. Overall, the study is well designed and the investigations carried out well. The data collection is complete and well described. However, there are several substantive issues that must be rectified.

  1. The biggest issue is conclusion by the authors that IL-8 levels, and IL-8 levels alone, are “independent” markers of disease progression and outcome. Their data show that IL-6 is also strong marker of disease progression and outcome. Their entire claim about IL-8 rests on statistical jugglery in Table 2 – something they call “multivariate analysis.” I require the numerical data for IL-6 to be included in Table 2. Also I require the authors to note that their own data show that IL-6 is singularly important(see Table 3). Also see the non-existent Kaplan Meir plot that should be Fig. 1. (its missing). Also please show IL-6 in Table 4. What emerges is that ignoring the statistical jugglery – both IL-6 and IL-8 are informative. Thus the authors must revise the MS to highlight IL-6 (in title, abstract and keywords).

All informations about the Tables 2 and 3 were corrected as follow: “The data emerged from multivariate analysis suggest that high systemic level of IL-8, but not IL-6, is associated with ESCC progression, despite the fact that IL-6 was highly significant for poor overall survival estimate by Kaplan-Meier”.

Actually, the IL-6 cytokine appears in the initial analysis as being statistically significant in relation to the survival of patients with esophageal cancer. However, subsequently, the variables that defined a value of p <0.2 were selected to compose the multivariate analysis using COX Regression (Table 4). Thus, only the IL-8 cytokine was statistically significantly lower, appearing to be a risk factor for the disease, since individuals who had increased levels of the cytokine had a greater risk of dying when compared to individuals with low levels of IL-8. Therefore, IL-6 values are not included in the table 2.

The requested Kaplan Meier plot is not supplemental material (Fig 2).

Based on your suggestions and comments we retitled the manuscript for  “Interleukin-8 and Interleukin-6 are biomarkers of poor prognosis in esophageal squamous cell carcinoma”.

  1. The second major problem is that various data items are missing. There is no Kaplan-Meir Figure 1. The various tables (especially in Supplement) are not in serial order as in the text. Fig. 2E in text is not in serial order with respect to Fig. 1. This makes it difficult for a reader to follow what the authors are saying.

The order was adjusted as recommended.

  1. There are many language issues: these are enumerated below.

25  delete “causisstry” and rephrase with “group” Done

27  delete “than” “compared to” Done

36  in keywords add IL-6 Done

45 delete “slightly” What does this mean? Rephrase has shown a small reduction

45  “decrease” What decrease? Previous sentence said increase and also slightly decrease. Rephrase. The phrase was modified.

  1. Rephrase “The etiology of ESCC is complex” wherein …the phrase was modified to The development of ESCC comprises a wide variety of etiological agents that may or may not act concomitantly.

52 delete “a” Done

52  its “Bhat” Done

53 delete “report” Done

54 change are to were – past tense Done

55 correklated past tense Done

59  change “in” to “on the” Done

60 change “in” to “on” Done

70 observations  plural Done

72 inducer Done

74 delete the Done

75 shown to be biomarker Done

84 change have to has Done

89 sduggests singular Done

109  what does “stipulated” mean. In Fig. S1 delete “included for convenience” Ha!! The text was modified

145 thawed past tense Done

163 suspensions plural Done

226 out of serial order Table S3. We modified the text following your suggestion. Thank you.

229  Table S4 not in serial order Done

241 change what to to We changed for Regarding

273 In Table 2 include IL-6 values Done

297-299 there is no Fig. 1 Kaplan Meir plot. We modified the text

Please show IL-6 in Table 4 Thank you for the suggestion. However, we decided to maintain the Table 4 as original be cause despite of significance of IL-6 as a promising iomarkers and considered statistically significant in the first analyses, after the multivariate analysis (where only variables with p<0.2 were included) IL-6 was not significant as presumed initially.

355 and 358 Fig 2E and Fig 1 are not in serial order. We modified the text following your suggestion. Thank you.

Ref. 21 is missing information. Information was added.

Round 2

Reviewer 1 Report

the author improved the manuscript

Reviewer 2 Report

The revisions are OK. The MS is much improved.